# Tuning the Molecular Structure of Corroles to Enhance the Antibacterial Photosensitizing Activity

**DOI:** 10.3390/pharmaceutics15020392

**Published:** 2023-01-24

**Authors:** Edwin J. Gonzalez Lopez, Sol R. Martínez, Virginia Aiassa, Sofía C. Santamarina, Rodrigo E. Domínguez, Edgardo N. Durantini, Daniel A. Heredia

**Affiliations:** 1IDAS-CONCIET-UNRC, Departamento de Química, Facultad de Ciencias Exactas Físico-Químicas y Naturales, Universidad Nacional de Río Cuarto, Agencia Postal Nro. 3, Río Cuarto X5804BYA, Argentina; 2IITEMA-CONICET, Departamento de Química, Facultad de Ciencias Exactas Físico-Químicas y Naturales, Universidad Nacional de Río Cuarto, Agencia Postal Nro. 3, Río Cuarto X5804BYA, Argentina; 3UNITEFA-CONICET, Departamento de Ciencias Farmacéuticas, Facultad de Ciencias Químicas, Universidad Nacional de Córdoba, Córdoba X5000HUA, Argentina; 4INFIQC-CONICET, Departamento de Química Orgánica, Facultad de Ciencias Químicas, Universidad Nacional de Córdoba, Córdoba X5000HUA, Argentina

**Keywords:** corrole, antimicrobial, reactive oxygen species, photosensitizers, photodynamic inactivation, super bugs, pH-activable cationic group

## Abstract

The increase in the antibiotic resistance of bacteria is a serious threat to public health. Photodynamic inactivation (PDI) of micro-organisms is a reliable antimicrobial therapy to treat a broad spectrum of complex infections. The development of new photosensitizers with suitable properties is a key factor to consider in the optimization of this therapy. In this sense, four corroles were designed to study how the number of cationic centers can influence the efficacy of antibacterial photodynamic treatments. First, 5,10,15-Tris(pentafluorophenyl)corrole (**Co**) and 5,15-bis(pentafluorophenyl)-10-(4-(trifluoromethyl)phenyl)corrole (**Co-CF_3_**) were synthesized, and then derivatized by nucleophilic aromatic substitution with 2-dimethylaminoethanol and 2-(dimethylamino)ethylamine, obtaining corroles **Co-3NMe_2_** and **Co-CF_3_-2NMe_2_**, respectively. The straightforward synthetic strategy gave rise to macrocycles with different numbers of tertiary amines that can acquire positive charges in an aqueous medium by protonation at physiological pH. Spectroscopic and photodynamic studies demonstrated that their properties as chromophores and photosensitizers were unaffected, regardless of the substituent groups on the periphery. All tetrapyrrolic macrocycles were able to produce reactive oxygen species (ROS) by both photodynamic mechanisms. Uptake experiments, the level of ROS produced in vitro, and PDI treatments mediated by these compounds were assessed against clinical strains: methicillin-resistant *Staphylococcus aureus* and *Klebsiella pneumoniae*. In vitro experiments indicated that the peripheral substitution significantly affected the uptake of the photosensitizers by microbes and, consequently, the photoinactivation performance. **Co-3NMe_2_** was the most effective in killing both Gram-positive and Gram-negative bacteria (inactivation > 99.99%). This work lays the foundations for the development of new corrole derivatives having pH-activable cationic groups and with plausible applications as effective broad-spectrum antimicrobial photosensitizers.

## 1. Introduction

The increase in bacterial resistance to antibiotics is one of the most severe problems that health care systems are facing nowadays. The mechanism by which bacteria develop resistance is much faster than the time that it takes to develop a new antibiotic [1,2]. Added to this, the low profits and lack of financial incentives have caused big pharmaceutical companies to give up on the antibiotic market [3]. This situation has led to the resurgence of new therapies to battle resistant pathogens. One of the most promising methodologies that has emerged to treat infections caused by resistant micro-organisms is photodynamic inactivation (PDI) [4]. This prominent approach, widely used to kill a broad spectrum of micro-organisms, requires the combination of a photosensitizer (PS), light, and triplet molecular oxygen (^3^O_2_) [5,6]. In PDI treatments, the irradiation of the PS, incorporated preferentially into microbial cells, and in the presence of ^3^O_2_, leads to the photogeneration of cytotoxic reactive oxygen species (ROS). The production of these species can be achieved via two competitive photodynamic mechanisms known as type I (production of free radicals) and type II (generation of singlet oxygen, ^1^O_2_) [7]. The cytotoxic ROS generated by both photomechanisms can non-specifically target biological substrates (such as proteins, lipids, nucleic acids, etc.), producing a loss of activity in these biomolecules, with the consequent cell death. In addition, the wide range of macromolecules attacked by ROS generates multiple and simultaneous cell damages, providing a broad spectrum of inhibitory activity and preventing pathogens from developing resistance against this therapy [8,9,10].

The main factor to enhance the antimicrobial photodynamic efficiency is directly related to the optimization of the physicochemical and photodynamic properties of PSs [11,12]. Thus, the obtention and characterization of new PSs for antimicrobial photodynamic therapy are of great relevance for the development of this area. Even though a wide variety of families of PSs have been applied to photokill pathogenic microbes [13,14,15,16,17,18], the most used are those based on porphyrins. The main reason relies on the high synthetic versatility of this kind of tetrapyrrolic macrocycles, which allows tuning the properties of PSs according to the needs. One disadvantage of porphyrins is their low absorption in the red region of the electromagnetic spectrum. This has led to continuous efforts to develop porphyrinoid derivatives with high red absorption [19]. On this point, the corrole core, a contracted isomer of porphyrin and the tetrapyrrolic constituent of vitamin B12 [19], is an exceptionally promising platform for the development of theragnostic agents for photodynamic therapy [20,21,22]. They present biocompatibility, efficient singlet oxygen generation, and non-cytotoxicity in the dark [22,23]. In addition, the versatile and facile synthetic methodologies to prepare and functionalize them, along with the co-ordination chemistry of this macrocycle, allow the designing of an unlimited number of derivatives by adjusting and improving their photochemical and structural properties. [24,25,26]. Consequently, corroles are outstanding PSs to be applied in phototreatments of pathogenic micro-organisms. Even so, only a few investigations have been carried out in the photoinactivation of microbes mediated by corroles. Moreover, there is a lack of knowledge concerning the relationship between the charge distribution in corroles and their capacity to photoinactivate bacteria. This encouraged us to study the effect of substituents of corrole derivatives on PDI efficiency.

In this work, we report the synthesis of four corrole derivatives that were properly functionalized to establish the relationships between the different substitution patterns in the molecular structure and their antimicrobial activity. The synthesis strategy allowed us to obtain four non-charged corroles. All compounds are neutral PSs; however, two of them are PSs with precursors of positive charges by protonation of the tertiary amines at physiological pH. The absorption and fluorescence properties of these macrocycles were determined and compared. The photodynamic behavior was investigated in the presence of different ROS scavengers. Photogeneration of ^1^O_2_ and superoxide anion radical (O_2_^•−^) were evaluated in solution. In addition, the levels of ROS produced in vitro were investigated using a fluorogenic reporter. On the other hand, the photosensitizing capability sensitized by these corroles was assessed in vitro against Gram-positive (methicillin-resistant *Staphylococcus aureus*, MRSA) and Gram-negative (*Klebsiella pneumoniae*) bacteria. Furthermore, the binding capacity of corroles to both bacteria was studied by Zeta potential measurements and compared with the PDI experiments. As far as we know, this is the first time that a PDI study has been performed using corroles that contain precursors of cationic groups (pH-activable) distributed in different substitution patterns. Our outcomes contribute and pave the way for the development of new corrole-based PSs and provide the best conditions for the photokilling of bacteria using these efficient broad-spectrum antimicrobial agents.

## 2. Materials and Methods

Instrumentation and supplies are described in Appendix A.

### 2.1. Synthesis of Corrole Derivatives

*Compound **DPM***: pyrrole (17.9 mmol, 1.23 mL) was added to an aqueous solution of HCl (0.18 M, 50 mL), then pentafluorobenzaldehyde (5.1 mmol, 1g) was added. The reaction mixture was stirred for 5 h at room temperature (r.t.) under argon atmosphere. After the indicated time, the white/orange semi-solid product, which often sticks to the walls and the stirring bar, was filtered off and washed with water and petroleum ether to give **DPM** (1.4 g, 4.48 mmol) in 88% yield. ^1^H NMR (400 MHz, CDCl_3_): δ 5.90 (1H, br s, CH), 6.03 (2H, br s, CH), 6.16–6.18 (2H, m, CH), 6.72–6.74 (2H, m, CH), 8.12 (2H, br s, NH). The spectroscopic data of **DPM** were in full agreement with those previously reported in the literature [27].

*Compound **Co***: To a solution of **DPM** (2.56 mmol, 800 mg) in methanol (MeOH, 100 mL), pentafluorobenzaldehyde (1.28 mmol, 251 mg) was added. Subsequently, an aqueous solution of HCl (5 mL of concentrated acid in 100 mL of distilled water) was added to the mixture and stirred for 1 h at room temperature under argon atmosphere. Afterward, the reaction mixture was extracted with dichloromethane (DCM), dried over anhydrous magnesium sulfate, and concentrated to 30 mL. This solution and a solution of 2,3-dichloro-5,6-dicyano-p-benzoquinone (DDQ, 2.64 mmol, 600 mg) in DCM/toluene (3/1, 16 mL) were simultaneously added to vigorously stirred DCM (30 mL). This mixture was stirred at 25 °C for 20 min, concentrated in vacuo to 15 mL, and passed through a short pad of silica, affording corrole with some impurities. The solvent was removed under reduced pressure, and the crude was purified by column chromatography (silica, DCM/petroleum ether, 1/9) to give **Co** (220 mg) in 22% yield. ^1^H NMR (500 MHz, CDCl_3_) δ 9.12 (d, *J* = 4.5 Hz, 1H), 8.77 (d, *J* = 4.5 Hz, 1H), 8.63–8.54 (m, 2H). ^19^F NMR (471 MHz, CDCl_3_) δ −137.18 (dd, *J* = 24.0, 8.5 Hz), −137.72 (d, *J* = 16.3 Hz), −152.19 (s), −152.76 (t, *J* = 20.8 Hz), −161.43 (s), −161.90 (ddd, *J* = 24.0, 20.1, 8.5 Hz). The spectroscopic data of **Co** were in full agreement with those previously reported in the literature [28].

*Compound **Co-CF_3_***: **DPM** (358 mg, 1.15 mmol) and 4-trifluoromethylbenzaldehyde (100 mg, 0.57 mmol) were dissolved in MeOH (80 mL). Subsequently, a solution of 4 mL concentrated HCl in 80 mL of water was added under argon atmosphere. The reaction mixture was stirred at room temperature for 1 h. The suspension was then extracted twice with DCM. The organic layers were combined, dried over anhydrous magnesium sulfate, and concentrated to 20 mL. This one and a solution of DDQ (3.45 mmol, 784 mg) in toluene/DCM (1:3, 14 mL) were simultaneously added to a vigorously stirred DCM (20 mL). The suspension was maintained at room temperature for 20 min and the reaction was concentrated in vacuo to 10 mL and passed through a short pad of silica. After that, the organic solvent was removed under reduced pressure and chromatographed (silica, DCM/petroleum ether, 1/9) to give **Co-CF_3_** (124 mg) in 28% yield. ^1^H NMR (300 MHz, CDCl_3_) δ 9.16–9.10 (m, 2H), 8.73 (d, *J* = 4.8 Hz, 2H), 8.64 (d, *J* = 4.8 Hz, 2H), 8.59 (d, *J* = 4.5 Hz, 2H), 8.31 (d, *J* = 7.9 Hz, 2H), 8.04 (d, *J* = 7.9 Hz, 2H). ^19^F NMR (282 MHz, CDCl_3_) δ −62.07, −137.86 (dd, *J* = 23.8, 8.3 Hz), −152.52 (t, *J* = 22.4 Hz), −161.25–−161.97 (m). ESI-MS [m/z] 774.1071 [M+H]^+^ (774.1089 calculated for [M+H]^+^, M = C_38_H_15_F_13_N_4_).

*Compound **Co-CF_3_-2NMe_2_***: *N*,*N*-dimethylethylenediamine (100 µL) was added to a stirred solution of **Co** (50 mg, 0.063 mmol) in dimethyl sulfoxide (DMSO, 4 mL). The reaction mixture was heated at 100 °C for 6 h under argon atmosphere. After that, DCM was added and the organic phase was washed with water to remove the excess of amine. The organic phase was dried over anhydrous magnesium sulfate and concentrated under reduced pressure. The residue was purified through column chromatography (silica, DCM/MeOH mixtures, 100:00 → 99:1) to afford **Co-CF_3_-2NMe_2_** (42 mg, 73%). ^1^H NMR (300 MHz, CDCl_3_) δ 9.06 (d, *J* = 4.3 Hz, 2H), 8.79 (d, *J* = 4.8 Hz, 2H), 8.60 (d, *J* = 4.3 Hz, 2H), 8.57 (d, *J* = 4.8 Hz, 2H), 8.31 (d, *J* = 7.8 Hz, 2H), 8.02 (d, *J* = 7.8 Hz, 2H), 3.71 (br. s, 4H), 2.71 (br. s, 4H), 2.38 (s, 12H). ^19^F NMR (282 MHz, CDCl_3_) δ −62.00, −141.91 (d, *J* = 14.5 Hz), −160.68 (d, *J* = 14.5 Hz). ESI-MS [m/z] 910.2923 [M+H]^+^ (910.2966 calculated for [M+H]^+^, M = C_46_H_37_F_11_N_8_).

*Compound **Co-3NMe_2_***: Cs_2_CO_3_ (245 mg, 0.756mmol) was added to a stirred solution of 2-Dimethylaminaethanol (200 µL) and **Co** (50 mg, 0.063 mmol) in DMF (2 mL). The reaction mixture was heated at 70 °C for 5 h under argon atmosphere. After that, DCM was added and the organic phase was washed with water to remove the excess of amine. The organic phase was dried over anhydrous magnesium sulfate and filtered. After removal of solvent under reduced pressure, the crude was chromatographed (silica, DCM/MeOH/TEA 9:0.8:0.2) to give **Co-3NMe_2_** (49 mg, 77%). ^1^H NMR (500 MHz, DMSO) δ 9.03 (d, *J* = 4.3 Hz, 2H), 8.64 (d, *J* = 4.7 Hz, 2H), 8.55 (d, *J* = 4.3 Hz, 2H), 8.46 (d, *J* = 4.7 Hz, 2H), 4.92 (q, *J* = 4.7 Hz, 6H), 3.70 (m, *J* = 4.7 Hz, 6H), 2.99 (s, 12H), 2.98 (s, 6H). ^19^F NMR (471 MHz, DMSO) δ −140.25 (t, *J* = 28.6 Hz), −157.38 (dd, *J* = 25.8, 8.6 Hz), −157.95 (dd, *J* = 26.1, 8.9 Hz). ESI-MS [m/z] 1004.3111 [M+H]^+^ (1004.3158 calculated for [M+H]^+^, M = C_49_H_42_F_12_N_7_O_3_).

### 2.2. Spectroscopic Characterization

Absorption and fluorescence emission spectra of corrole derivatives were recorded in diluted toluene solutions at room temperature and using a cell quartz of 1 cm path length. For fluorescence emission spectra, **Co**, **Co-CF_3_**, **Co-CF_3_-2NMe_2_**, and **Co-3NMe_2_** were absorbance (A < 0.05) matched at the excitation wavelength (λ_exc_= 572 nm). Samples were excited at 572 nm and fluorescence emission was detected between 600 and 800 nm. Fluorescence quantum yields (Φ_F_) were calculated according to previously published work (see Appendix A) [18,29], using **Co** as a reference (Φ_F_ = 0.10 in toluene) [30]. Fluorescence excitation spectra were recorded between 360 and 700 nm at an emission wavelength of 710 nm.

### 2.3. Singlet Oxygen Detection

Singlet oxygen photosensitized by corrole derivatives was detected according to previous works [18,29]. Samples, containing 9,10-dimethylantracene (DMA) and the corresponding corrole (A = 0.1 at 606 nm) in toluene (2 mL), were irradiated with light (λ_irr_ = 606 nm, 1.15 mW/cm^2^) under aerobic conditions. The photo-oxidation of DMA followed a pseudo first-order kinetic and was monitored by the decrease of its absorption band at λ_max_^DMA^ = 379 nm. The observed rate constants (*k_obs_*) were calculated by plotting the natural log of the absorbance at 379 nm at time *t* divided by the initial absorbance (A_0_) vs. time (s) to give *k_obs_* as the slope of the linear fit of the data. Quantum yields of ^1^O_2_ (Φ_Δ_) production, measured under the same conditions, were calculated by comparing the *k_obs_* of each corrole with that of the reference (**Co**, Φ_Δ_ = 0.58 [30]) according to Appendix A [18,29].

### 2.4. Superoxide Anion Radical (O_2_^•−^) Detection

The O_2_^•−^ photogeneration was determined by measuring the increase of the absorbance of diformazan (λ = 560 nm) formed by the reduction of nitro blue tetrazolium (NBT) in the presence of reduced nicotinamide adenine dinucleotide (NADH). Samples containing NADH (0.5 mM), NBT (0.2 mM), and the corresponding corrole derivative (A = 0.1 at 628 nm) in air-equilibrated *N*,*N*-dimethylformamide (DMF)/water (9/1, 2 mL) were irradiated with light (λ_irr_ = 628 nm, 1.18 mW/cm^2^). Control experiment was performed by irradiating a corrole-free solution.

### 2.5. Photodynamic Inactivation in Planktonic Cultures

Cultures were stored in glycerol 10% (*v*/*v*) and Tryptic Soy Broth (TSB) 90% (*v*/*v*) at −80 °C. Clinical isolates of the bacterial strains (MRSA 771 and *K. pneumoniae*) were grown in Tryptic Soy Agar (TSA) at 35 °C for 18 h, after which bacterial culture was prepared by inoculating one single isolated colony from a pure culture in TBS. The bacterial inoculum was transferred to fresh broth and incubated until the optical density (OD_600_) reached a value of ~0.2. Finally, the bacterial suspension is 100-fold diluted in phosphate-buffered saline (PBS) at pH 7.2 to achieve a bacterial concentration of ~10^6^ colony-forming units (CFU)/mL according to the McFarland standard. Afterward, a stock solution of each corrole was prepared in DMF (1 × 10^−4^ M). Then, 195 µL of bacterial suspension was incubated with 5 µL of each stock solution in DMF (final concentration 2.5 µM). Later, the plate was kept in the dark for 20 min. Then, the samples were illuminated with a fluent incident light power of 90 mW/cm^2^. Aliquots (20 µL) of control and irradiated suspensions were 10-fold serially diluted with PBS. Each dilution was plated (10 µL) TSA, the number of colony-forming units per milliliter (CFU/mL) was enumerated after 18–24 h incubation, and the CFU/mL was log-transformed. Samples were run in triplicate [31].

The minimum inhibitory concentration (MIC) for the four compounds was tested according to the Clinical and Laboratory Standards Institute guidelines with some modifications [32]. The corrole compounds affect per se the bacterial viability when the cells are treated with concentrations above 10 µM.

The corrole derivatives were stable in the stock solutions kept at 4 °C for 40 days. These solutions were used in the biological experiments and no significant differences were found in the antimicrobial action after 40 days.

The aqueous stability of the corroles was spectroscopically monitored. Solutions of the macrocycles in PBS were left in a closed quartz cuvette (10 × 10 mm) for 8 days. The solutions were kept in the dark at room temperature. UV-visible absorption spectra were recorded each day. Appendix A shows representative results of the experiment. As shown in Appendix A, a slow and slight decomposition of the corrole core was observed after 8 days.

### 2.6. Bacterial Uptake by Zeta Potential Studies

The Zeta potential (ζ) is defined as the electric charge at the shear plane. We took advantage of this methodology to measure the adsorption/uptake of the molecules within the bacterial structures. Cells were harvested in TBS overnight and then centrifuged 3 times at 3500 rpm and suspended in PBS to reach a final concentration of ~10^8^ CFU/mL. Samples were incubated with corroles (2.5 µM) for 20 min at room temperature and kept in the dark. Then, the ζ of the samples was determined by photon correlation spectroscopy (dynamic light scattering, DLS) and electrophoretic light scattering (ELS) measurements, using a Zetasizer Nano-S instrument (Malvern Instruments^®^, Malvern, UK). The pH of the buffer solution was 7.4.

### 2.7. Reactive Oxygen Species Production upon Light Therapy

Bacterial suspensions were grown aerobically in TS broth (OD 600 nm ~1). Fresh samples were centrifuged for 10 min at 4000 rpm, and the pellets were rinsed two times and suspended in PBS. Stock solutions of corrole were prepared in DMF (1 × 10^−4^ M); 195 μL of bacterial suspensions were incubated with 5 μL of each compound for 20 min in the dark. After that, samples were irradiated for 20 min (90 mW/cm^2^). Finally, 20 μL of the fluorescence reporter 2′,7′-dichlorodihydrofluorescein diacetate (DCFH_2_-DA) were added to each well (final concentration of the probe 1 µM) and incubated for 30 min in a dark environment at 35 °C. DCFH_2_-DA was excited at 490 nm and the fluorescence emission was collected at 520 nm using a multi-mode microplate reader, Biotek [33].

## 3. Results and Discussion

### 3.1. Design and Synthesis of Corrole Derivatives

The increase in microbial resistance has led to the search for new and more efficient PSs. Therefore, it is vital to recognize how structural features affect the antimicrobial efficiency. It is well-established that the position and number of positive charges in the PS structure play a significant role in the photokilling action [34,35,36]. Thus, with this in mind and motivated by the scarce knowledge about this topic in corroles derivatives, we focused on the design and synthesis of corroles with different substitution patterns to study the relationship between the molecular structure and its performance as PSs to photoinactivate pathogens.

The synthetic strategy to prepare the tetrapyrrolic macrocycles is outlined in Figure 1. First, the synthesis required the previous preparation of **DPM**. For this step, we used a “greener” methodology reported by Dehaen and collaborators, which has several advantages compared to the classical Lindsey conditions [37]. The *meso*-substituted dipyrromethane was obtained from the acid-catalyzed (HCl) condensation of three equivalents of pyrrole (**1**) with pentafluorobenzaldehyde (**2**) in water at room temperature. The precipitated 5-pentafluorophenyldipyrromethane was filtered off and washed to afford in high yield (88%) the desired dipyrrolic building block. Under these conditions, the use of trifluoroacetic acid, excess of pyrrole (solvent), and purification by chromatography, all characteristics of Lindsey’s conditions, were avoided.

After that, the aryldipyrromethane was subjected to a condensation reaction by treating it with the corresponding aromatic aldehydes (pentafluorobenzaldehyde and 4-(trifluoromethyl)benzaldehyde) in a mixture of methanol/aqueous HCl, to afford bilane derivatives. Then, subsequent oxidation with DDQ at room temperature gave rise to corroles **Co** and **Co-CF_3_** as green solids in 22 and 28% yields, respectively. These compounds are versatile building blocks to construct more elaborate PSs since they can be derivatized through the highly regioselective nucleophilic aromatic substitution reaction (S_N_Ar) [38,39,40]. The latter is characterized by the displacement of the *para*-fluoro atoms and has high reaction yields [36,41]. In addition, perfluorophenyl (PFP) units could also be used for ^19^F magnetic resonance imaging [42,43], which, combined with the fluorescence emission of the corroles, turns them into interesting theragnostic agents with potential biomedical applications.

Having secured an efficient route to synthesize the macrocycles, the next task was directed toward the functionalization of the tetrapyrrolic cores with basic aliphatic amino groups. These substituents will act as precursors of cationic centers by protonation at phycological pH [36,44]. Initial attempts to subject both corroles to S_N_Ar reactions using 2-(dimethylamino)ethylamine as a nucleophile only afforded the *meso*-amino-substituted corrole **Co-CF_3_-2NMe_2_** in 73% yield [45]. On the other hand, the incorporation of the precursors of positive charges on **Co** under these conditions was unsuccessful, and only unreacted starting material and multiple TLC spots corresponding to different degrees of substitution were observed (see Image S1). Unfortunately, all attempts to improve the reaction conditions, such as other solvents (DMSO, DMF, and THF), temperatures (25, 60, and 100 °C) and number of equivalents (three, five, and eight equivalents) met with no success, obtaining the same previous outcomes. Thus, keeping in mind to introduce external basic amine groups, we decided to use 2-dimethylaminaethanol as nucleophile and Cs_2_CO_3_ as base in DMF, which cleanly afforded **Co-CF_3_-2NMe_2_** in 77% yield.

All the tetrapyrrolic macrocycles were characterized by nuclear magnetic resonance (NMR) (Appendix A). The high regioselectivity of the S_N_Ar reactions was confirmed by ^19^F NMR spectra. The substituted products showed the complete disappearance of resonances corresponding to the *para*-fluorine atoms (δ ~ 152–154 ppm). The signal of *meta*-fluorine atoms, which are sensitive to the molecular structure of the nucleophile, gave rise to a doublet around δ ~ 160 ppm, while *orto*-fluorine atoms appeared around δ ~ 140 ppm. Signals corresponding to -CF_3_ unit for **Co-CF_3_** and **Co-CF_3_-2NMe_2_** were observed around δ ~ 60 ppm as singlets. Concerning the ^1^H NMR spectra, **Co-3NMe_2_** combined the diagnostic resonances of the corrole macrocycle in the aromatic region (β-pyrrolic protons) together with characteristic resonances corresponding to nucleophile moiety. **Co-CF_3_** and **Co-CF_3_-2NMe_2_** showed, in the downfield region, a splitting of the β-pyrrolic signals as a result of the asymmetry of these *trans*-A_2_B-corroles. In all cases, integrals of alkyl signals corresponding to the (CH_3_)_2_NCH_2_CH_2_- fragment demonstrated the incorporation of the nucleophile and confirmed the complete replacement of the *para*-fluor atoms in the PFP groups.

At this point, we have four corrole derivatives. **Co** and **Co-CF_3_** have two PFP units at positions 5 and 15, and differ from each other in the aryl substituent at position 10: **Co** and **Co-CF_3_** have PFP and trifluoromethylphenyl moieties, respectively. This trifluoromethylphenyl group at the *meso* position was introduced to achieve a different degree of substitution. Thus, **Co-3NMe_2_** and **Co-CF_3_-2NMe_2_** were obtained with three and two precursors of cationic centers, respectively. The peripheral basic *N*,*N*-dimethylamino groups are protonated at physiological pH, acquiring positive charges on the N atom of the amine units.

The combination of these precursors of positive charges with the lipophilic fluorinated macrocycle confers an amphiphilic character to the molecular structures. The cationic centers interact with the overall negatively charged bacterial cell wall, while the hydrophobic nature of the fluorinated moieties interacts with the lipid membrane [46,47]. Furthermore, cationic groups not only improve the association and incorporation of the PS to micro-organisms, which increase the photocytotoxic activity, but also enhance the aqueous solubility by the interaction of the positive charges with water. It is worthwhile to note that the tertiary amino groups are bonded through an alkyl chain, giving them higher mobility to interact with micro-organisms [36,47,48]. In addition, this non-conjugated aliphatic spacer leads to generate localized positive charges exposed in the outer periphery of the corroles with a negligible effect on their photophysical properties. Thus, it is expected that molecular structural features of **Co-3NMe_2_** and **Co-CF_3_-2NMe_2_** gave rise to highly effective broad-spectrum antimicrobial agents.

### 3.2. Spectroscopic Properties

Absorption and fluorescence properties of the corrole derivatives were determined and compared in toluene-diluted solutions. Figure 1a shows the absorption spectra of the four tetrapyrrolic macrocycles in toluene at room temperature, and Table 1 summarizes their main spectroscopic characteristics. As shown in Figure 1a, the four aromatic organic compounds presented similar absorption features, which are characteristics of corroles. All the examined tetrapyrrolic macrocycles showed typical Soret bands around 420 nm and Q bands in the region 500–660 nm [30]. The maxima of the Soret bands corresponding to the non-amino-substituted corroles, which have molar absorptivity coefficients (ε) of ~10^5^ M^−1^cm^−1^, exhibited a small red-shift compared to **Co-3NMe_2_** and **Co-CF_3_-2NMe_2_**. The observed bathochromic shifts result from the auxochromic effect induced by the oxy and amino substituents. Regarding the Q bands, with lower molar absorptivity, this effect is less noticeable when **Co**/**Co-3NMe_2_** and **Co-CF_3_**/**Co-CF_3_-2NMe_2_** are compared. The absorption studies demonstrate that the peripheral derivatization does not significantly influence electronic transitions of the corrole-based chromophores.

Fluorescence steady-state emission spectroscopy, performed in diluted toluene solutions, was carried out by exciting the samples at 572 nm (Figure 1b). As can be observed from the spectra, all compounds showed the distinctive red emission profile of corroles, with a main fluorescence band around 655 nm and a shoulder centered at ~720 nm [30]. These emission bands correspond to the transitions that take place from the first singlet excited level to the first two vibrational transitions of the ground state, named *Q*(0–0) and *Q*(0–1). The main band and the shoulder correspond to the 0→0 and 0→1 vibrational transition of the S1 → S0 electronic transition, respectively. As shown in Table 1, the emission maxima wavelengths exhibited a red-shift depending upon the molecular structure and are consistent with the bathochromic shift of the Q-bands in the absorption spectra. The obtained data indicated similar excited-state behavior for these macrocycles in toluene. The values of Φ_F_ of corroles **Co-CF_3_**, **Co-3NMe_2_**, and **Co-CF_3_-2NMe_2_** were determined by a comparison with **Co**, which was used as a reference compound [30]. All compounds displayed similar values in the range of 0.10–0.15, which are in accordance with values reported in the literature for these kinds of macrocycles. Their adequate fluorescence properties turn them into promising probes to investigate intracellular localization or quantify the PS in biological media through fluorescence microscopy studies [20,48,49].

Finally, excitation spectra of corroles were recorded in toluene at room temperature (Appendix A) by monitoring the emission at 710 nm. This spectroscopic technique is highly sensitive and can be very useful to quantify and differentiate the absorption bands of corroles, especially when they are present in very low concentrations or forming aggregates. In addition, fluorescence excitation spectroscopy provides valuable insight into the absorption bands when corroles are dissolved in biological media with other organic compounds of similar absorption features. The four corroles presented similar fluorescence excitation spectra, which also resemble their corresponding absorption spectra. This fact indicated that corroles are mainly found as monomers (non-aggregated) in the solvent used.

### 3.3. ROS Photosensitized by Corroles

ROS production can be achieved by two photoreactions known as type I and type II (Appendix A). In the first one, the excited triplet-state PS (^3^PS^*^), produced by light absorption, reacts with biological substrates by electron transfer processes giving rise to highly oxidizing radicals, which in the presence of molecular oxygen produce O_2_^•−^. This ROS can then dismute to hydrogen peroxide, precursor of hydroxyl radicals. Alternatively, in type II photochemical reaction, the ^3^PS^*^can directly transfer energy to molecular oxygen leading to the production of ^1^O_2_.

A key feature to be considered in the photodynamic properties of a PS is its capability to produce ^1^O_2_. This highly oxidizing species can non-specifically react with an extensive range of biomacromolecules. Therefore, the capability of the corroles to photosensitize ^1^O_2_ was assessed by an indirect method, using DMA as a scavenger of this ROS. DMA quenches ^1^O_2_ mainly by chemical reaction, generating an endoperoxide product (see Appendix A). Thus, the photo-oxidation of the chemical trap can be monitored by UV–visible spectroscopy, recording the decrease of DMA absorption band at 379 nm. Hence, DMA photodecomposition mediated by all corroles was carried out upon irradiation at λ_irr_ = 606 nm in toluene under aerobic conditions.

The spectroscopic changes of DMA as a function of the irradiation time are shown in Appendix A. The absorption bands of DMA decreased gradually under illumination in the presence of **Co**, **Co-CF_3_**, **Co-3NMe_2_**, and **Co-CF_3_-2NMe_2_**, evidencing that ^1^O_2_ was produced by the photosensitizing action of all derivatives. Further data from Appendix A shows that Soret and Q bands remained unchanged throughout the experiment, indicating that the tetrapyrrolic macrocycles were photostable and retained their spectroscopic features. It is worth mentioning that no changes in DMA spectra were detected in the presence of the corroles in the dark.

Figure 2 demonstrates a pseudo-first-order kinetic behavior (as described by Appendix A) and similar rates of ^1^O_2_ photosensitization for the corrole derivatives. The slope of the linear fitting of ln(A_0_/A) vs. time gave rise to the values of *k_obs_^DMA^*, which are gathered in Table 1. The *k_obs_^DMA^* values for **Co-CF_3_**, **Co-3NMe_2_**, and **Co-CF_3_-2NMe_2_** were compared with that for the reference (corrole **Co**) in order to calculate Φ_Δ_ values, which presented similar yields for all of them. Furthermore, they agree with those previously reported for this family of macrocycles. It is worth highlighting that the different substitution patterns do not remarkably influence the type II mechanism, and all corroles can directly undergo an energy transfer process to molecular oxygen, giving ^1^O_2_.

Even though ^1^O_2_ generation is a key parameter to determine the efficiency of PSs, type I pathway also plays an important role in the photoinactivation process [50]. Thus, in order to determine cytotoxic reactive species generated by a type I process, NBT method was used to prove the formation of O_2_^•−^. NBT reacts with this ROS generating diformazan, a reduced state of the molecular probe (see Appendix A) [51,52]. The electron transfer reaction is favored by polar environments and the presence of a biological reducing agent like NADH.

Solutions containing corrole, NBT, and NADH in DMF/H_2_O were irradiated at 628 nm under aerobiosis. Moreover, a solution of NBT and NADH, without corrole, was used as a control. The reduction of the tramping agent was spectroscopically followed by the increase of the absorption band of diformazan at λ = 560 nm. Figure 3 shows spectral changes corresponding to the generation of difomarzan as a function of the irradiation time. As can be observed, all corroles exhibited higher values than NBT + NADH control, which means that the four macrocycles can sensitize the NBT reduction. However, from the comparisons among the curves in Figure 3, it is clear that these PSs are not efficient producers of O_2_^•−^ under these experimental conditions. It is well established that PSs with good capability to accept electrons are more susceptible to undergo a type I mechanism [53], and tetrapyrrolic macrocycles predominantly participate in type II reactions [54].

Despite the presumable low production (NBT method is not quantitative), **Co**, **Co-CF_3_**, **Co-3NMe_2_**, and **Co-CF_3_-2NMe_2_** are able to produce O_2_^•−^ after irradiation. This oxidant specie presents significant relevance since it can dismute to H_2_O_2_, which is converted by another reduction in the harmful oxidant hydroxyl radical HO^•^, a process known as the Fenton reaction [55].

Based on the studies of the photodynamic properties, our PSs can generate ROS through the two photodynamic pathways, regardless of their external functionalization on the macrocycle. Nevertheless, these results found in solution are not extrapolated to the cellular environment since several factors, such as localization and interaction of the PSs with the microenvironment, polarity, etc., can affect the photodynamic properties of the corroles.

### 3.4. Antimicrobial Photodynamic Therapy

Uptake of the corrole derivatives by bacteria is a key feature for an efficient antimicrobial action. Consequently, before assessing the PDI treatments mediated by the corroles, information about the uptake of **Co**, **Co-CF_3_**, **Co-3NMe_2_**, and **Co-CF_3_-2NMe_2_** to MRSA and *K. pneumoniae* was assessed. The capability of the corrole to bind to both bacterial structures was studied in PBS (cell density ~1 × 10^8^ CFU/mL). The bacterial uptake was determined using ζ measurements. Table 2 summarizes the results of both bacterial samples incubated for 20 min with each compound. **Co-3NMe_2_** and **Co-CF_3_-2NMe_2_** that were protonated at pH 7.4 demonstrated better adsorption/incorporation to MRSA and *K. pneumoniae*. In addition, **Co-3NMe_2_**, the molecule that showed the best performance in PDI, is the one that changes ζ to more positive values. This trend is followed by **Co-CF_3_-2NMe_2_** (Table 2). In contrast, the neutral molecules did not change the overall surface charge of the bacterial samples.

After that, the four compounds were tested in MRSA and *K. pneumoniae.* Both strains came from clinical isolates and were evaluated in planktonic suspensions. The samples were illuminated for 30 min with a 90 mW/cm^2^ potency and dark controls were run in parallel. Bacterial survival was not affected for samples kept in dark (see Appendix A). Whereas bacterial viability was highly compromised when both strains were treated with **Co-3NMe_2_**, a bactericidal effect was attained meaning a drop in 99.999% of cell samples upon light treatment (Figure 4, dark grey column). **Co-CF_3_-2NMe_2_** showed a killing efficiency of ~99% (Figure 4, dark cyan column). The compounds that were not protonated at physiological pH displayed a ~2.5 drop of log in CFU/mL—**Co** and **Co-CF_3_**, the light grey and light cyan columns, respectively. It is worth highlighting that the results of killing profile efficiency after light treatment are in accordance with the bacterial uptake. On the other hand, the American Society of Microbiology established that a compound has to reach an inactivation efficiency higher than 99.9% to be defined as antimicrobial/antibacterial [56]. Therefore, since **Co-3NMe_2_** exhibited a photokilling higher than 99.9% in both bacteria, it can be considered as one.

Finally, the levels of ROS elicited were studied using the fluorogenic reporter H_2_DCF-DA to evaluate the impact of the oxidative burst in both bacterial structures. Figure 5 illustrates the induction of oxidative species when samples were exposed solely to light or light plus each studied molecule. Green bars display the basal levels of intracellular ROS production, while grey and cyan bars show bacteria treated with compounds + light. It is easy to visualize the substantial rise in oxidant species when samples were exposed to positively charged complexes (Figure 5, dark grey and dark cyan bars). A slight increase is appreciated in the case of cells treated with neutral compounds.

Herein, the results demonstrated that positive charge molecules are better photodynamic agents than neutral or anionic compounds. In this sense, we and others have evidenced this fact on pathogenic strains of different species [31,36,57,58]. Moreover, cationic structures have been shown to be more efficient in killing robust Gram-negative bacteria [59,60]. Notably, we proved that **Co-3NMe_2_** have better uptake than **Co-CF_3_-2NMe_2_** in both bacterial structures. Halder et al. reported that surface-acting agents produced changes of zeta potential, in Gram-positive and Gram-negative strains, and those could be translated into enhanced surface permeability [61]. Likewise, our results with both protonated structures at physiological pH evidenced a shift in zeta potential to more positive values upon incubation with bacteria. The bacterial uptake results are directly reflected on the killing efficiency of cationic structures with respect to the neutral compounds.

Recently, corrole structures have been reported for photodynamic therapy of various types of cancer cells [21,23,62]. In addition, these chemical structures were tested in mold fungi spores [63]. The authors demonstrated the ability of the compound to photosensitize ^1^O_2_ and carry out a complete inactivation of spores with positively charged complexes. In addition, previous reports of porphyrin derivates, which combined highly lipophilic groups (PFP and CF_3_) and precursors of positive charges, have also demonstrated that these amphiphilic structures are highly effective broad-spectrum antimicrobial PSs [47,64,65]. To the best of our knowledge, this is the first report on using corroles to treat fastidious pathogens such as MRSA and *K. pneumoniae*. Moreover, we confirmed the ROS induction upon light treatment of both pathogens, pointing out that the oxidative damage is the main cause of cell death.

## 4. Conclusions

Four corroles bearing different substitution patterns were designed and prepared through a short and straightforward synthetic sequence. All of them are neutral PS; however, two derivates contain different numbers of aliphatic tertiary amine groups, which are precursors of cationic centers by protonation at physiological pH.

Spectroscopic and photodynamic studies demonstrated that the corrole-based PSs retained their properties regardless of the derivatization at the periphery of the macrocycle. All PSs exhibited the characteristic absorption and fluorescence properties of the corrole ring. Furthermore, these macrocycles were able to generate ROS by both photodynamic mechanisms. Thus, the four corroles have an adequate balance between ROS production and fluorescence emission, turning them into theragnostic agents for PDI.

Even though the different external substitution did not alter their physicochemical features, a remarkable impact on the binding to micro-organisms and their antimicrobial action was observed, as was shown by studies of ζ and PDI treatments. Molecular structures with precursors of positive charges were better uptaken by both Gram-positive and Gram-negative bacteria. Furthermore, in vitro PDI treatments over both clinical strains showed higher antimicrobial activity for those derivates with protonatable tertiary amines. **Co-3NMe_2_** was able to eliminate both strains when cultures were incubated with 2.5 µM PS and irradiated for 30 min, turning it into a promising PS despite not having intrinsic cationic centers in its molecular structure. These experimental conditions, low concentration of PS and low fluence of white light, are appropriate for antimicrobial photodynamic therapy.

Our results proved that the external substitution pattern strongly impacts the antimicrobial activity. In addition, it was demonstrated that the number of peripheral basic amino groups improves the interaction with negatively charged cell envelopes. As far as we know, this is the first time that non-charged corrole PSs bearing precursors of positive charges are applied in PDI. Thus, we demonstrated that the design of corroles for photodynamic applications should consider this structural feature. This research paves the way for the development of new corrole-based PSs, producing a great impulse in the use of this family of macrocycles for PDI applications.

## Data Availability

Not applicable.

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
