# Peer review of "Tuning the Molecular Structure of Corroles to Enhance the Antibacterial Photosensitizing Activity"

_pharmaceutics, 2023, doi:10.3390/pharmaceutics15020392_

Round 1

Author Response

Replies to Reviewer #1

Comment #1. Ln 86, I think that” The synthetic strategy” would be better if replaced with “The synthesis strategy”.

Response. Thank you very much for your comment to improve our work. The sentence was modified as the reviewer requested.

Comment #2. From the sentence in Ln 469 something is missing. please rewrite.

Response. Thank you very much for your observation. In agreement with the reviewer’s advice, the sentence was corrected.

Comment #3. In ln 498 punctuation should be revised.

Response. Thank you very much for your comment. The punctuation was revised and modified.

Comment #4. In the introduction a sentence regarding the obtaining and characterization of some new photosensitizers for antimicrobial photodynamic therapy could be introduced.

Response. Thank you very much for your suggestion to improve our work. In agreement with the reviewer’s opinion, a new sentence was introduced in the Introduction section.

Reviewer 2 Report

The bacterial resistance to antibiotics is a serious threat to public health. Photodynamic inactivation is one of the most promising methods to treat infections related with bacteria. In this manuscript, the authors report the synthesis of four corrole-based photosensitizer derivatives and investigate the relationships between the different substitution patterns in the molecular structure and their antimicrobial activity. The manuscript is well-written & organized but some points must be explained before publication.

1. Whether the PS activity is related to the water solubility of the compound? The authors should provide some data about the water solubility of four compounds.

2. Please explain the “HRMS:” in line 164 of the manuscript.

3. How to monitor and adjust the power density of lamp? The authors describe that “samples were irradiated for 20 min (90 mW/cm2)” in the text. However, a Cole-Parmer illuminator 41720-series produces a light fluence rate of 1.15 mW cm-2 at 606 nm ± 5 nm and 1.18 mW cm-2 at 628 nm ± 5 nm. Which type of light source is used to irradiate the bacteria? Please explain it.

Author Response

General Comment. The bacterial resistance to antibiotics is a serious threat to public health. Photodynamic inactivation is one of the most promising methods to treat infections related with bacteria. In this manuscript, the authors report the synthesis of four corrole-based photosensitizer derivatives and investigate the relationships between the different substitution patterns in the molecular structure and their antimicrobial activity. The manuscript is well-written & organized but some points must be explained before publication.

Response. Thank you very much for your positive judgment of our work. We did our best in the design and characterization of new antimicrobial photosensitizers based on corroles.

Comment #1. Whether the PS activity is related to the water solubility of the compound? The authors should provide some data about the water solubility of four compounds.

Response. Thank you very much for your comment to improve our work. It is an interesting point and we understand the reviewer's concern; however, all compounds were soluble at the working concentration. It could be possible that compounds containing precursors of positive charges (Co-CF3-2NMe2 and Co-3NMe2) present a better interaction with water molecules. Despite all this, the antimicrobial activity is a consequence of the interaction of positive charges with the negatively charged cell envelopes of the studied strains.

Comment #2. Please explain the “HRMS:” in line 164 of the manuscript.

Response. Thank you very much for your observation. In agreement with the reviewer’s comment, the word “HRMS” was deleted.

Comment #3. How to monitor and adjust the power density of lamp? The authors describe that “samples were irradiated for 20 min (90 mW/cm2)” in the text. However, a Cole-Parmer illuminator 41720-series produces a light fluence rate of 1.15 mW cm-2 at 606 nm ± 5 nm and 1.18 mW cm-2 at 628 nm ± 5 nm. Which type of light source is used to irradiate the bacteria? Please explain it.

Response. Thank you very much for your careful observation. Experiments in cell suspensions were carried out with a Novamat 130 AF slide projector containing a 150 W 24 V halogen lamp with a power density of 90 mW/cm2.

Reviewer 3 Report

This manuscript reported the synthesis of four appropriately functionalized corrole derivatives to establish the relationship between different substitution patterns in the molecular structure and their antimicrobial activity. The results of this work are of importance for the development of new corrole derivatives having pH-activable cationic groups and with plausible applications as effective broad-spectrum antimicrobial photosensitizers. Overall, I would like to recommend its major revision, and the following issues should be addressed.

The detailed comments are as follows:

1. Zeta potential changes alone cannot accurately characterize the bacterial uptake of compounds. I think corresponding experiments should be supplemented, such as the use of confocal microscopy to film the uptake of compounds by bacteria.

2. Figure 2, 3 shows similar rates of 1O2 photosensitization for the corrole derivatives, and these PSs are no efficient producers of O2•−. However, Figure 5 shows that there is a large amount of intracellular ROS production upon PDI in Co-3NMe2 group. In addition to 1O2 and O2•−, are other types of ROS produced? If so, please experimentally confirm the presence of other types of ROS.

3. The toxicity to normal cells is not known, which is the most critical point in photosensitizer applications.

4. It was mentioned that the better bacterial uptake ability of the cationic structure compared with the neutral compound led to the stronger bacterial killing efficiency, so what is the antibacterial mechanism involved?

5. Why use a corrole derivatives concentration of 2.5 μM for in vitro antimicrobial photodynamic therapy experiments? Is it the optimal antibacterial concentration? I believe that different concentrations of corrole derivatives should be supplemented for their cytotoxicity and photodynamic antimicrobial effects.

6. It is recommended to check references to ensure uniform format, such as ref. 40, 44, 64.

Author Response

General Comment. This manuscript reported the synthesis of four appropriately functionalized corrole derivatives to establish the relationship between different substitution patterns in the molecular structure and their antimicrobial activity. The results of this work are of importance for the development of new corrole derivatives having pH-activable cationic groups and with plausible applications as effective broad-spectrum antimicrobial photosensitizers. Overall, I would like to recommend its major revision, and the following issues should be addressed.

Response. Thank you very much for your feedback and suggestions. We did our best in the design, synthesis and characterization of new antimicrobial photosensitizers based on corroles.

Comment #1. Zeta potential changes alone cannot accurately characterize the bacterial uptake of compounds. I think corresponding experiments should be supplemented, such as the use of confocal microscopy to film the uptake of compounds by bacteria.

Response. Thank you very much for your helpful advice. We consider that it is a very useful measurement, but unfortunately, we currently do not have the availability to perform confocal microscopy experiments at our workplace.

Comment #2. Figure 2, 3 shows similar rates of 1O2 photosensitization for the corrole derivatives, and these PSs are no efficient producers of O2•−. However, Figure 5 shows that there is a large amount of intracellular ROS production upon PDI in Co-3NMe2 group. In addition to 1O2 and O2•−, are other types of ROS produced? If so, please experimentally confirm the presence of other types of ROS.

Response. Thank you very much for your careful observation. The main pathway of the photodynamic mechanism (type I and type II) was determined in solution. However, this can change significantly in a cellular medium, depending on the polarity of the microenvironment and the availability of substrates where the corrole is located. Therefore, it is difficult to directly correlate the photodynamic data obtained in solution with those in bacterial suspension. Other ROS, such as H2O2, OH., and peroxyl radicals, could be formed; however, we do not have the reagents to determine the formation of these ROS in cellular environments. 

Comment #3. The toxicity to normal cells is not known, which is the most critical point in photosensitizer applications.

Response. Thank you very much for your opinion regarding the toxicity of photosensitizers on eukaryotic cells. Corrole derivatives have been largely studied as photosensitizer molecules to treat cancer cells (doi.org/10.1021/acs.chemrev.6b00400). In this sense, authors have demonstrated on different cell lines that a wide variety of these molecules do not show toxicity when the cell lines are incubated with the corrole derivative for up to 72 hours in dark conditions (doi.org/10.1073/pnas.151740211, doi.org/10.1016/j.jinorgbio.2014.06.015).

Furthermore, recently in 2022, Barata and co-workers have demonstrated on Vero cell lines that two corrole derivatives affect cell viability from 11.7 to 23.3 % with respect to the untreated control. Thus, the ISO guidelines point out that for a molecule to be considered safe for clinical applications, less than 30 % of the cell viability can be affected over the studied period (doi.org/10.3390/microorganisms10061167). On the other hand, the binding of photosensitizers to bacteria is a faster process than that into mammalian cells.

At our home University, we do not have the facilities to evaluate the cytotoxicity of the compounds on mammalian cells. However, since the bacterial uptake times are shorter than eukaryotic cells, we support the present results with the evidence that colleagues in the field have previously demonstrated.

Comment #4. It was mentioned that the better bacterial uptake ability of the cationic structure compared with the neutral compound led to the stronger bacterial killing efficiency, so what is the antibacterial mechanism involved?

Response. Thank you very much for your comment. The antibacterial mechanism involved is the ROS production, which damages multiple biomacromolecules of the cellular system producing irreversible changes and leading to cell death. These ROS are produced by the irradiation of the photosensitizer in the presence of molecular oxygen. 

As can be observed in the dark controls, the survival of the microbial pathogens was not affected by irradiation with visible light in the absence of corroles. In addition, methicillin-resistant Staphylococcus aureus and Klebsiella pneumoniae strains treated with 10 μM of corroles in the dark did not show alteration in cell viability. As expected, the photosensitizing action of all corroles took place after visible light irradiation. In all cases, the viability of both bacteria was significantly reduced compared to the control. Nevertheless, higher antimicrobial activities were found for those corroles containing precursor of positive charges. It is well known that the bacterial wall acts as a barrier to the uptake of photosensitizers, and the photosensitizer-cell wall interaction is directly associated with the molecular structural features of this one. Positively charged phototherapeutic agents establish a strong interaction with the negative sites on the outer bacterial cell wall, favoring the binding to the cell envelope. This is essential to improve the photocytotoxicity, increasing the photokilling of pathogenic microorganisms. Thus, the outcomes can be explained and supported through the molecular structure of the corroles. All corroles derivatives are non-charged photosensitizers. However, at pH = 7.2, the tertiary basic amino groups in Co-3NMe2 and Co-CF3-2NMe2 can be protonated, acquiring positive charges. Consequently, the presence of these cationic centers leads to an enhancement of the binding to the cell envelope, increasing the antimicrobial action.

Comment #5. Why use a corrole derivatives concentration of 2.5 μM for in vitro antimicrobial photodynamic therapy experiments? Is it the optimal antibacterial concentration? I believe that different concentrations of corrole derivatives should be supplemented for their cytotoxicity and photodynamic antimicrobial effects.

Response. Thank you very much for your comment. The minimum inhibitory concentration (MIC) for these compounds was measured. This methodology allows studying the toxicity of different concentrations of the corroles when they are incubated with ~ 10 6 cells. We assessed concentrations between 20 µM to 0.015 µM for each compound, and after 24 h of incubation, the turbidity (optical density) of the samples was corroborated. The MIC of the four compounds was 10 µM. The MIC methodology has an error associated with ± 2-fold dilutions. For this reason, 2.5 µM was selected to evaluate the efficiency of the compounds as PS. A new paragraph was included in the manuscript. 

Comment #6. It is recommended to check references to ensure uniform format, such as ref. 40, 44, 64.

Response. Thank you very much for your observation to improve our work. The format of the references was checked and corrected.

Reviewer 4 Report

The authors presented research aimed at obtaining new corrole derivatives that were properly functionalized to establish the relationships between the different substitution patterns in the molecular structure and their antimicrobial activity, especially against clinical, resistant bacterial strains.

I understand that the novelty of the work is the use of uncharged  photosensitizers containing precursors of positive charges in photodynamic inactivation.

Authors should clearly indicate the novelty of the work in relation to the available works in the field of photodynamic inactivators.

In my opinion, the authors should present cytotoxicity studies if the synthesized compounds are dedicated as potential effective  phototherapeutics.

Have the authors studied stability? This question is very important from the point of view of potential applications (especially medical applications) and should be examined.

Author Response

Comment #1. Authors should clearly indicate the novelty of the work in relation to the available works in the field of photodynamic inactivators.

Response. Thank you very much for your comment. One of the main factors to enhance the antimicrobial photodynamic action is directly related to the optimization of the photosensitizer structure. Despite the numerous investigations with porphyrinoids in PDI found in the literature, very few reports have been carried out using corroles. Moreover, there is a lack of knowledge concerning the relationship between the different substitution patterns in corroles with their capacity to photoinactivate bacteria.

A large number of phototherapeutic agents based on tetrapyrrolic macrocycles can be found in the literature. However, our work expands the family of corrole-based photosensitizers. In addition, many investigations can be found using photosensitizers with intrinsic positive charges to photokill pathogenic microorganisms. Here, we introduced, for the first time, new corrole-based therapeutics agents derivatized with precursors of cationic centers by protonation of their tertiary amines at physiological pH. The dyes are characterized by intense red absorption, high generation of singlet molecular oxygen, suitable stability, and insignificant cytotoxicity in the dark. The advantages of these tetrapyrrolic macrocycles have been mentioned through the main text.

In addition, here we demonstrated the easy synthetic derivatization of the corrole core, which enables the fine-tuning of its molecular structure. Thus, we are positive that these derivatives have a bright future since the quest for new supramolecular structures with suitable properties for applications in PDI depends only on the researcher’s inspiration.

Comment #2. In my opinion, the authors should present cytotoxicity studies if the synthesized compounds are dedicated as potential effective phototherapeutics.

Response. Thank you very much for your opinion regarding the toxicity of photosensitizers on eukaryotic cells. Corrole derivatives have been largely studied as photosensitizer molecules to treat cancer cells (doi.org/10.1021/acs.chemrev.6b00400). In this sense, authors have demonstrated on different cell lines that a wide variety of these molecules do not show toxicity when the cell lines are incubated with the corrole derivative for up to 72 hours in dark conditions (doi.org/10.1073/pnas.151740211, doi.org/10.1016/j.jinorgbio.2014.06.015).

Furthermore, recently in 2022, Barata and co-workers have demonstrated on Vero cell lines that two corrole derivatives affect cell viability from 11.7 to 23.3 % with respect to the untreated control. Thus, the ISO guidelines point out that for a molecule to be considered safe for clinical applications, less than 30 % of the cell viability can be affected over the studied period (doi.org/10.3390/microorganisms10061167). On the other hand, the binding of photosensitizers to bacteria is a faster process than that into mammalian cells.

At our home University, we do not have the facilities to evaluate the cytotoxicity of the compounds on mammalian cells. However, since the bacterial uptake times are shorter than eukaryotic cells, we support the present results with the evidence that colleagues in the field have previously demonstrated.

Comment #3. Have the authors studied stability? This question is very important from the point of view of potential applications (especially medical applications) and should be examined

Response. Thank you very much for your comment. In agreement with the reviewer’s observation, we consider that the decomposition of the corrole unit is a key aspect to be considered. Therefore, we added the requested experiment in the experimental section and the supporting information. The stability of the corrole in PBS was spectroscopically monitored. Solutions of the PSs in PBS were left in a closed quartz cuvette (1 × 1 cm) for 8 days. The sample was kept in the dark at room temperature. Absorption spectra were acquired each day. A slow and slight decomposition of the corrole unit was observed after 8 days. In addition, it is worth mentioning that the corrole derivatives were stable in DMF solutions kept at 4°C for 40 days. These stock solutions were used in the biological experiments and no significant differences were found in the antimicrobial action after 40 days. Two paragraphs and one graph were added to the manuscript and the supplementary information, respectively.

Round 2

Reviewer 3 Report

Revised manuscript satisfy this journal.

Reviewer 4 Report

The manuscript has been sufficiently improved. The authors provided suggested tests/explanations. Thank you for your cooperation.